# Autoantibodies to ACE2 and immune molecules are associated with COVID-19 disease severity
Eric S. Geanes[1], Rebecca McLennan [1], Cas LeMaster [1] & Todd Bradley [1,2,3,4] ✉

## Abstract

**Background** Increased inflammation caused by SARS-CoV-2 infection can lead to severe coronavirus disease 2019 (COVID-19) and long-term disease manifestations. The mechanisms of this variable long-term immune activation are poorly defined. One feature of this increased inflammation is elevated levels of proinflammatory cytokines and chemokines. Autoantibodies targeting immune factors such as cytokines, as well as the viral host cell receptor, angiotensin-converting enzyme 2 (ACE2), have been observed after SARS-CoV-2 infection. Autoantibodies to immune factors and ACE2 could interfere with normal immune regulation and lead to increased inflammation, severe COVID-19, and long-term complications.

**Methods** Here, we deeply profiled the features of ACE2, cytokine, and chemokine autoantibodies in samples from patients recovering from severe COVID-19. We measured the levels of immunoglobulin subclasses (IgG, IgA, IgM) in the peripheral blood against ACE2 and 23 cytokines and other immune molecules. We then utilized an ACE2 peptide microarray to map the linear epitopes targeted by ACE2 autoantibodies.

**Results** We demonstrate that ACE2 autoantibody levels are increased in individuals with severe COVID-19 compared with those with mild infection or no prior infection. We identify epitopes near the catalytic domain of ACE2 targeted by these antibodies. Levels of autoantibodies targeting ACE2 and other immune factors could serve as determinants of COVID-19 disease severity, and represent a natural immunoregulatory mechanism in response to viral infection.

**Conclusions** These results demonstrate that SARS-CoV-2 infection can increase autoantibody levels to ACE2 and other immune factors. The levels of these autoantibodies are associated with COVID-19 disease severity.

## Plain language summary

Antibodies are small proteins that are produced by your immune system to protect you when an unwanted foreign invader such as bacteria, viruses and toxins enters your body. When these antibodies target proteins on our own cells instead of the invader, we call them autoantibodies. Autoantibodies that target host immune molecules, as well as ACE2, a receptor molecule that interacts with the SARS-CoV-2 virus, have been observed after COVID-19. We found that patients who had severe COVID-19 displayed higher levels of these autoantibodies compared to those who had mild infection or were uninfected. These findings suggest that these autoantibody levels could serve as indicators of COVID-19 severity.

The exact causes of severe COVID-19, or the long-lasting effects of SARS-CoV-2 infection, are not yet well defined. One possible contributing mechanism could be the persistent overactivation of the immune system, driving inflammation and cellular injury and disrupting host tissue homeostasis[1–3]. While inflammatory signaling is critical for viral clearance, heightened or prolonged inflammation could lead to increased tissue damage and adverse disease outcomes.

Acute cases of COVID-19 and associated disease outcomes have been shown to correlate with elevated expression levels of proinflammatory cytokines and chemokines[4,5]. The enzyme angiotensin-converting enzyme 2 (ACE2) is not only the host viral receptor for SARS-CoV-2 but also plays a key role in the renin–angiotensin system that can regulate systemic and local inflammation[6–8]. ACE2 levels have been shown to be inversely correlated with markers of inflammation, and mice with genetic knock-out of ACE2

[1]Genomic Medicine Center, Children's Mercy Research Institute, Kansas City, MO, USA. [2]Department of Pediatrics, University of Missouri, Kansas City, MO, USA. [3]Department of Pediatrics, University of Kansas Medical Center, Kansas City, KS, USA. [4]Department of Pathology and Laboratory Medicine, University of Kansas Medical Center, Kansas City, KS, USA. ✉e-mail: tcbradley@cmh.edu

present a hyperinflammation phenotype[6–9]. It has been reported that SARS-CoV-2 infection can directly alter ACE2 levels, leading to increased inflammation[10]. In addition to altering ACE2 levels and elevated levels of inflammation, individuals with prior SARS-CoV-2 infections produce autoantibodies targeting ACE2, type I interferons, and other immune molecules. These autoantibodies have been associated with more severe COVID-19 disease outcomes[11–20]. Interestingly, increased autoantibodies to cytokines or other autoantigens have also been observed in other respiratory infections and critical illnesses involving inflammation[21]. Thus, the generation of autoantibodies to proinflammatory immune molecules, including ACE2, could represent a common immunoregulatory mechanism for controlling inflammation. In the case of COVID-19, investigating the levels and activity of ACE2, as well as the creation of autoantibodies, could provide insight into the extent of inflammatory responses and disease severity outcomes. Utilizing these profiles may provide a more accurate understanding of how to treat and predict COVID-19 and other respiratory disease severity prior to infection. There have been reports that both IgG and IgM isotypes of ACE2 autoantibodies are associated with COVID-19 disease severity[13,22]. However, the characterization of the features and functions of ACE2 autoantibodies and their impact on COVID-19 disease outcomes within the same cohort has not yet been described.

In this study, we find that autoantibodies targeting ACE2 and other immune factors are higher in individuals with severe COVID-19. Using high-resolution epitope mapping, we identify an immunodominant epitope near important residues for ACE2 substrate binding and enzymatic activity. Levels of autoantibodies targeting ACE2 and other immune factors may serve as determinants of COVID-19 disease severity and represent an important immunoregulatory mechanism after viral infection.

## Methods

### Human participants

Healthcare workers from our children's hospital were enrolled prior to the administration of the Comirnaty® mRNA COVID-19 vaccine (Pfizer-BioNTech, New York, NY, USA). Plasma from peripheral blood was collected before vaccination as a baseline (week 0) and after the second immunization (week 7) from individuals with no known history of infection ($n = 38$) or with PCR laboratory-confirmed SARS-CoV-2 infection and no hospitalization ($n = 33$). The sample population consisted of mostly adult middle-aged, white females who did not identify as Hispanic or Latino (Supplementary Table 1). Comirnaty® vaccine biospecimens were collected under a research study at Children's Mercy Kansas City. This study was reviewed and approved by the Children's Mercy IRB (#00001670 and #00001317). Participants self-enrolled after they had reviewed a study information letter and were given the opportunity to ask questions. IRB waived written informed consent after these criteria were met.

Severe COVID-19 convalescent biospecimens, 30–60 days after infection, were obtained through Boca Biolistics, LLC (Pompano Beach, FL, USA). Informed written consent was obtained from individuals with acute or convalescent SARS-CoV-2 infection to obtain biospecimens for prospectively collected COVID-19 research study. The samples were collected under a clinical study that has been reviewed by an Institutional/Independent Review Board (IRB) and/or Independent Ethics Committee (IEC) in accordance with requirements of local governing regulatory agencies, including the Department of Health and Human Services (DHHS) and Food and Drug Administration (FDA) Codes of Federal Regulations, on the Protection of Human Subjects (45 CFR Part 46 and 2l CFR Part 56, respectively). The biospecimens were deidentified, and key de-linked demographic (e.g., age, sex) and clinical (e.g., COVID-19 status) were provided with each biospecimen to Children's Mercy Investigators. The use of these deidentified biospecimens as nonhuman subjects research for the purpose of this study was reviewed and approved by the Children's Mercy IRB (Supplementary Table 1). These convalescent individuals had PCR laboratory-confirmed SARS-CoV-2 infection and were hospitalized. Serum or plasma was isolated from venous whole blood collection and stored

frozen in ultra-low temperature freezers until used to perform immunoassays.

### ACE2 enzyme-linked immunosorbent assays

ELISAs were performed using ACE2 recombinant protein (Cat# HCY-TAAB-17K, Sino Biological, Wayne, PA, USA) diluted to 2 µg/mL in 0.1 M sodium bicarbonate and incubated on high-binding plates (3369, Corning Inc., Corning, NY, USA) overnight at 4°. Serum or plasma was diluted to 1:50 in superblock buffer with sodium azide. Secondary antibodies were purchased from the following: Goat anti-human IgG (Cat# 109-036-098, Lot# 149163, Jackson ImmunoResearch, West Grove, PA, USA), Goat Anti-Human IgA alpha chain (Cat# ab97215, Lot# GR3373878-8, Abcam, Waltham, MA, USA), Goat Anti-Human IgM mu chain (Cat# ab97205, Lot# GR3396429-1, Abcam, Waltham, MA, USA). Secondary antibody dilutions were done in superblock buffer without sodium azide within range of manufacturer's recommendations at: IgG 1:50,000, IgA 1:1000, and IgM 1:2000 dilution. SureBlue Reserve Microwell Substrate (95059-294, VWR, Radnor, PA, USA) was added and incubated in the dark for 15 minutes. Absorbance was measured at 450 nm immediately after 0.33 N HCl Acid Stop solution was added to the plate. Positive baseline cutoff was determined by values greater than or equal to twice the background OD450.

### ACE2 peptide microarray

A peptide library of 15 amino acids that overlapped by 11 amino acids (199 total peptides) that spanned the entire ACE2 protein were synthesized by JPT peptide Technologies (Berlin, Germany) using PepStar technology that covalently immobilized the peptides onto glass microarray surfaces using an optimized hydrophilic linker moiety. Full-length human and mouse IgG were co-immobilized on microarray slides as assay controls. The 20 severe COVID-19 serum samples were diluted 1:200 and incubated for 1 hour at 30 °C on multiwell microarray slides. After incubation and washing, fluorescently labeled anti-human-IgG antibody at 0.1 µg/mol was added to the wells and incubated for 1 h. Additional control incubations with secondary antibody only (with no serum samples) were also performed in parallel on each slide to assess false positives. After washing and drying, the slide was scanned with a high-resolution laser scanner (GenePix; Molecular Devices, San Jose, CA, USA) at 635 nm to obtain fluorescence intensity profiles. The resulting images were quantified to yield a mean pixel value for each peptide. The blocking buffer was Superblock TBS T20 (Pierce International), and the wash buffer was 50 mM TBS buffer, including 0.1% Tween20, pH 7.2. To visualize epitopes on the structure of ACE2, we analyzed the location of the immunodominant linear antibody epitopes on the previously published crystal structure of ACE2 in complex with the SAR-CoV-2 receptor-binding domain (RCSB PDB:6M0J)[23]. For structure visualization and manipulation, we used EzMol version 2.1[24] on the downloaded structure coordinates from the PDB[25].

### Autoantibody multiplexed binding assay

To measure autoantibody levels against cytokines commonly associated with autoantibody disease, 23 cytokines were used on a bead-based multiplex assay based on the Luminex xMAP technology (Austin, TX, USA). Reagent kits with secondary antibodies specific for immunoglobulin G, A or M (IgG/A/M) were used (HCYTAAB-17K, HCYTABA/M/G, MilliporeSigma, Burlington, MA, USA) following manufacture protocol. The kit provided a set of 23 antigen-conjugated beads (BAFF/Blys, G-CSF, IFNβ, IFNγ, IL-1α, IL-6, IL-8, IL-10, IL-12 (p40), IL-15, IL-17A, IL-17F, IL-18, IL-22, TNFα, IL-4, IFNω, GM-CSF, IL-3, IL-2, IFNα2, Osteopontin, PF-4) along with 3 positive control beads and a negative control bead set. The positive control beads were beads coated with different concentrations of IgG/A/M. The negative control beads did not have antigens conjugated to determine nonspecific binding. The 23 antigen-conjugated beads, 3 positive control beads, and 1 negative control beads were mixed and incubated with each plasma sample at a dilution of 1:100 with assay buffer. Two wells with only buffer and no plasma were used to determine background activity. PE-anti-human IgG/A/M conjugate detection antibody was utilized to

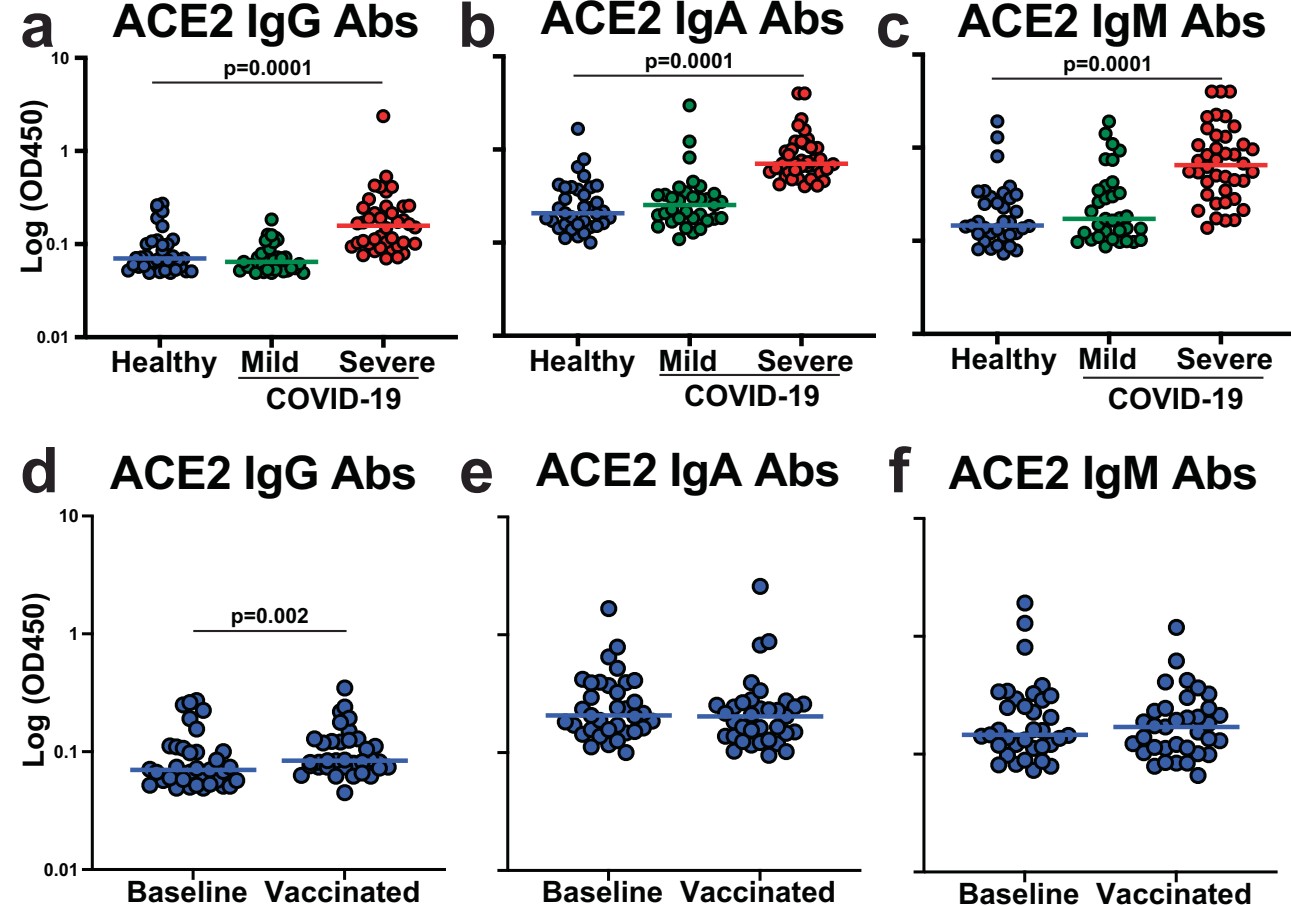

**Fig. 1 | ACE2 autoantibody levels are increased in severe COVID-19. a–f** Dot plot graphs of OD450 values on the log axis obtained by ELISA for determining the levels of antibodies targeting ACE2 within plasma from healthy individuals ($n = 35$, blue), individuals with mild COVID-19 (not hospitalized, $n = 33$, green), and individuals with severe COVID-19 (hospitalized, $n = 40$, red). Secondary detection antibodies specific for Ig isotype were utilized to identify IgG ACE2 antibody levels **b** IgA antibody levels, and **c** IgM antibody levels. **d–f** Dot plots showing ACE2 antibody levels in healthy individuals at baseline and then after two doses of Comirnaty COVID-19 mRNA vaccination. Ig isotype-specific ACE2 antibodies determined for **d** IgG **e** IgA and **f** IgM. Each dot represents a single individual, and Wilcoxon–Mann–Whitney statistical test was used to determine *P* values.

determine antibody response to each cytokine. We utilized the Luminex analyzer (MAGPIX) and Luminex xPONENT acquisition software to acquire and analyze data. After the acquisition, net MFI was calculated by subtracting background MFI (no plasma).

### Statistics and reproducibility
The statistical analysis was performed using GraphPad Prism 9.1 (Boston, MA, USA). For multiple comparisons, the statistical significance was determined with a Wilcoxon–Mann–Whitney test with two-tailed *P* values. Heatmaps were generated using Morpheus (Broad Institute), and visualization of ACE2 and SARS-CoV-2 spike crystal structure and residue annotation was performed using EzMol version 2.1[24].

### Reporting summary
Further information on research design is available in the Nature Portfolio Reporting Summary linked to this article.

## Results
### Elevated ACE2 autoantibodies in individuals recovered from severe COVID-19
We used plasma samples from healthy individuals with no prior SARS-CoV-2 infection ($n = 38$) and from individuals 30–60 days after mild ($n = 33$; defined as no hospitalization) or severe ($n = 40$; defined as requiring hospitalization) COVID-19 disease (Supplementary Table 1). We measured

the levels of antibodies that targeted ACE2 using immunoglobulin (Ig) isotype-specific ELISAs. First, we used plasma samples from healthy individuals and determined individuals that were considered positive for ACE2 autoantibodies using a positivity cut-off of twice the background reading for each Ig isotype (Supplementary Fig. 1). We did not have access to isotype-specific ACE2 autoantibody standards so the levels reported are comparing the intensity of the colorimetric substrate measured as optical density. The highest overall OD450 levels of ACE2 autoantibodies were observed for IgA isotype, followed by IgM and IgG. We found 12 IgG, 16 IgA, and 17 IgM ACE2 autoantibody positive individuals of the 35 healthy individuals tested: indicating the presence of preexisting anti-ACE2 antibodies within healthy individuals (Supplementary Fig. 1). Next, we compared the levels observed in the healthy individuals to those with mild or severe COVID-19. There were no significant differences in the levels of ACE2 autoantibodies of any isotype when comparing healthy individuals to individuals who had recovered from mild COVID-19 (Fig. 1a–c). However, individuals with severe COVID-19 had significantly higher levels of ACE2 autoantibodies compared to healthy individuals for all three ACE2 Ig isotypes (IgG, IgA, and IgM) (Fig. 1a–c). We also found a significant correlation between IgA and IgM ACE2 autoantibodies in the individuals with severe COVID-19, but not with IgG and IgA or IgM subclasses (Supplementary Fig. 2). These data demonstrated that ACE2 autoantibodies for all Ig subclasses could be detected, and individuals hospitalized with severe

**Fig. 2 | High-resolution IgG antibody epitope mapping using ACE2 peptide microarray identified regions of targeted binding in individuals with severe COVID-19. a** Schematic of ACE2 protein over peptide regions with known domains and motifs highlighted. **b** Line graphs of median group z-scores of ACE2 autoantibody binding to individual ACE2 peptides. Epitopes of recurrent high binding were qualified as median z-scores of ≥1 to represent 1 standard deviation above the median. Seven regions qualified as high binding (grey; labeled peptide ID **a–g**). **c** Heatmap of z-scores for ACE2 epitope binding for individual samples, n = 20.

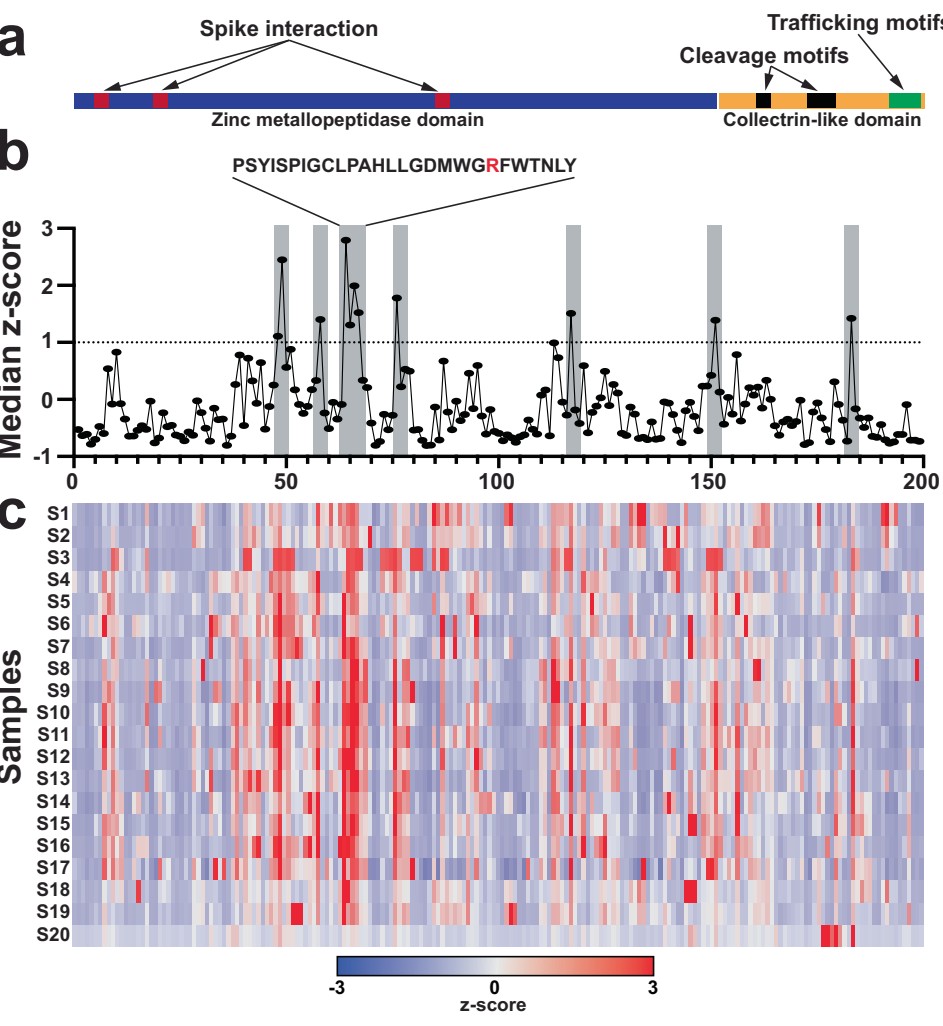

COVID-19 had higher levels of ACE2 autoantibodies after recovery compared to those that were healthy or had mild disease.

We next determined if COVID-19 vaccination altered the levels of ACE2 autoantibodies in healthy individuals after two doses of the Comirnaty® mRNA COVID-19 vaccine. After vaccination, there were no significant differences in ACE2 IgA or IgM autoantibody levels compared to levels before vaccination (Fig. 1e, f). There was a statistically significant (P = 0.002; Wilcoxon-paired test) increase in the IgG levels after two doses of the vaccine, but the magnitude of the change was very small (1.2 fold-change of the median), and the 95% CI of the baseline ranged from 0.06 to 0.10 and after vaccination ranged 0.08–0.19, indicating only a small increase in magnitude (Fig. 1d). Thus, vaccination with a COVID-19 mRNA vaccine had little effect on the magnitude of ACE2 autoantibodies in healthy individuals.

**Autoantibodies from individuals with severe COVID-19 targeted immunodominant epitopes in ACE2**

To map the high-resolution antibody epitopes targeted by the ACE2 IgG autoantibodies in individuals with severe COVID-19, we utilized a peptide microarray that spanned the full-length ACE2 protein that consisted of a peptide library of 15 amino acids that overlapped by 11 amino acids (199 total peptides). We selected plasma from 20 individuals with severe COVID-19 containing the highest levels of ACE2 IgG autoantibodies detected in our ELISA assays to epitope map using these peptide arrays (Supplementary Data 1). To visualize results, we used z-scores that were computed based on the individual peptide fluorescence intensities for each individual across all the peptides that spanned ACE2 displayed as a

heatmap. ACE2 consists of two major domains, a zinc metallopeptidase domain (blue) and a collectrin-like domain (yellow) (Fig. 2a). The motifs required for substrate binding, enzymatic activity, and interaction with the SARS-CoV-2 spike protein (red) are contained in the zinc metallopeptidase domain[26–29]. The collectrin-like domain contains cleavage sites for ADAM18, TMPRSS11D, and TMPRSS2 peptidases (black) and several linear binding motifs for LIR, PDZ-binding, PTB, and endocytic sorting signal motifs (green) that are important for trafficking and autophagy[30–32] (Fig. 2a). Antibody binding to peptides across the entirety of the ACE2 protein occurred throughout the array of individuals. However, there were common epitopes targeted by multiple samples. To quantify this, we calculated the median z-scores for each peptide across all 20 individuals and defined a z-score of ≥1 (representing 1 standard deviation above the median) as high binding (Fig. 2b (gray), 2c). Seven regions met this criterion, with only two regions containing more than one peptide with high binding (Fig. 2b, Table 1). None of the high binding regions overlapped with the regions of ACE2 previously shown to be important for interacting with SARS-CoV-2[26,27,30,31]. One of the regions with the highest binding did involve residues important for substrate binding (Fig. 2b, c). Mutation of the arginine at residue 273 has been shown to block substrate binding and abolish enzymatic activity when mutated[28].

**Epitopes targeted by ACE2 autoantibodies are near regions important for ACE2 function**

We further analyzed the location of the immunodominant linear antibody epitopes on the crystal structure of ACE2 in complex with the SAR-CoV-2 receptor-binding domain (RBD; PDB:6M0J; Table 1)[23,25]. The SARS-CoV-2

spike RBD binding to the ACE2 protein is independent of ACE2 catalytic activity and occurs on the outer surface of the molecule[29,33–35] (Fig. 3a). One of the peptide epitopes (peptide ID: g) was in the C-terminal collectrin-like domain and not resolved in this crystal structure. Four other peptide regions (peptide ID: b, d, e, f) bind to the outside of the ACE2 protein but not near the SARS-CoV-2 RBD interaction site or the cleft important for ACE2 catalytic activity (Fig. 3a). Two of the immunodominant peptide epitopes are located in the ACE2 cleft important for substrate binding and enzymatic activity (peptide ID: a, c; Fig. 3a). Prior structural analysis of an ACE2 catalytic inhibitor (MLN-4760) identified key residues important for ACE2 substrate binding and carboxypeptidase activity[29]. The two peptide regions targeted by antibodies in multiple individuals (peptide id: a, c) overlap with these regions previously shown to be important for ACE2 substrate binding and subsequent enzyme activity (Fig. 3b). These data showed that ACE2 autoantibodies had common linear epitopes across many individuals and some antibody epitopes targeted regions that may be important for ACE2 enzymatic activity.

### Detection of autoantibodies for cytokines, chemokines, and other immune factors of various Ig subclasses

In addition to autoantibodies targeting ACE2, autoantibodies to cytokines, chemokines, and other immune molecules have also been associated with COVID-19 disease severity[14,15,17–19]. Therefore, we utilized a bead-based fluorescence Luminex assay to determine the levels of autoantibodies against 23 cytokines, chemokines, and immune molecules for IgG, IgA, and IgM isotypes. We found high levels of autoantibodies were present in certain cytokines in the healthy individuals, but most analytes displayed large levels

of heterogeneity (Fig. 4a). IgG autoantibodies against IL-4, IL-17F, IL-17A, IFNω, and IFNγ had the highest median levels in the healthy group (Fig. 4a). IgM levels were significantly higher than most analytes compared with IgG, with the exception of IL-4, IL-15, and IL-22 where IgG exhibited significantly higher levels (Fig. 4b). Conversely, IgA levels to all the analytes, except osteopontin, were significantly lower than IgG levels (Fig. 4c). Thus, autoantibodies to cytokines, chemokines, and other immune factors could be detected in healthy individuals at different magnitudes, depending on the individual and analyte. Moreover, there were distinct patterns based on Ig isotype.

### Individuals with severe COVID-19 had higher IgG and IgA autoantibodies to immune factors

Next, we compared the patterns of autoantibodies in the severe COVID-19 individuals to the healthy controls. For the IgG autoantibodies, the severe COVID-19 group trended for higher levels in many of the analytes tested and had statistically significant higher levels of both IL-3 and IFNα2 cytokines ($p \leq 0.05$; Wilcoxon–Mann–Whitney; Fig. 5a, Supplementary Fig. 3). Similarly, the severe COVID-19 group had higher levels of IgA autoantibodies to most cytokines with 18 of the 23 analytes significantly higher in the severe COVID-19 group compared to healthy controls ($p \leq 0.05$; Wilcoxon–Mann–Whitney; Fig. 5b, Supplementary Fig. 3). There were no significant differences in IgM autoantibody levels between severe COVID-19 and healthy groups (Supplementary Fig. 3). Subsequently, we determined if any cytokine autoantibody levels were correlated with one another in the IgG isotype. Interestingly, there were groups of correlating analytes in the severe COVID-19 group that were not detected in the healthy group (Fig. 5c). In the severe COVID-19 group, levels of autoantibodies against IL-1α, IFNα2, TNFα, osteopontin and IFNβ all showed high correlation (Fig. 5c). These data identified that autoantibodies to cytokines, chemokines, and other immune factors were present in healthy individuals at varying degrees of magnitude and are dependent on Ig isotype. Moreover, individuals with severe COVID-19 had significantly higher levels of autoantibodies targeting cytokines and immune factors for both IgG and IgA, but not IgM, immunoglobulin subclasses.

### Discussion

In this study, we showed that autoantibodies of all major Ig subclasses (IgG, IgA, IgM) targeting the SARS-CoV-2 viral host cell receptor ACE2 were present in the blood, and the levels were associated with COVID-19 disease severity. We found that individuals hospitalized with COVID-19 had higher ACE2 autoantibody levels of all three Ig isotypes. This correlation of ACE2 antibodies with disease severity is consistent with previous studies, but this is

**Table 1 | Peptide sequences and amino acid location on ACE2 protein for peptides that had antibody binding across multiple individuals (z-score ≥ 1; immunodominant)**

| Peptide ID | Peptide sequence | ACE2 AA sequence number |
|---|---|---|
| a (48-49) | EMARANHYEDYGDYWRGDY | 189–207 |
| b (58) | TFEEIKPLYEHLHAY | 229–243 |
| c (64-67) | PSYISPIGCLPAHLLGDMWGRFWTNLY | 253–279 |
| d (76) | AWDAQRIFKEAEKFF | 301–315 |
| e (117) | KGEIPKDQWMKKWWE | 465–479 |
| f (151) | NSFVGWSTDWSPYAD | 601–615 |
| g (183) | PTLGPPNQPPVSIWL | 729–743 |

**Fig. 3 | Mapping of immunodominant linear antibody epitopes on ACE2 complexed with SARS-CoV-2 receptor-binding domain molecular structure. a** Crystal structure of the SARS-CoV-2 spike receptor-binding domain bound with ACE2 (PDB: 6M0J)[23,25]. ACE2 protein in dark green, RBD in orange, immunodominant peptide regions shaded red and labeled with peptide ID. **b** Closer view of the ACE2 enzyme catalytic site with residues found bound to an ACE2 enzyme drug inhibitor (MLN-4760) shaded in purple.

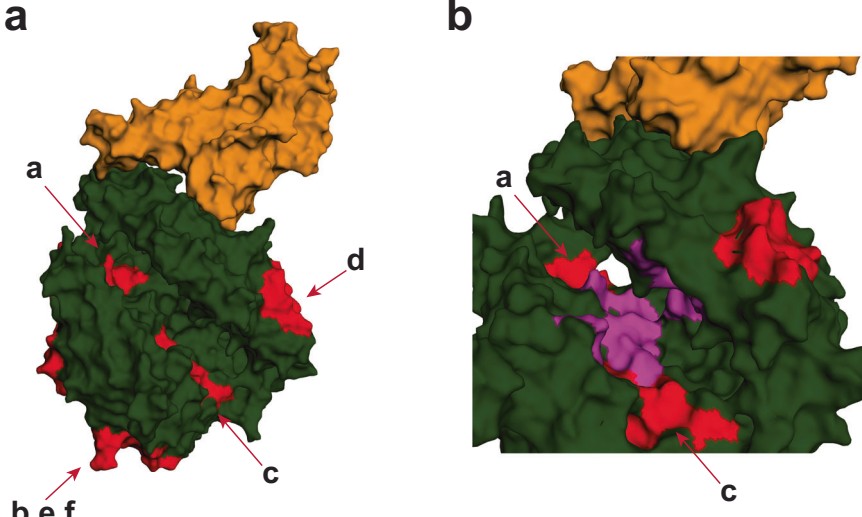

**Fig. 4 | Autoantibodies for cytokines, chemokines, and other immune factors were detected for different Ig isotypes within healthy individuals.**
**a** Multiplex bead-based antibody binding assay measured the levels of autoantibodies against 23 cytokines, chemokines, and immune molecules for IgG (blue), IgA (red), and IgM (green) isotypes within healthy individual plasma samples ($n = 38$). Median fluorescent intensity (MFI) was calculated; background subtraction was used to remove non-specific signals. Box and whisker plots are shown. The box shows the lower and upper quartile, with the median indicated by the solid line. The lines extending from the box (whiskers) indicate the upper and lower extreme values. The dashed line indicates a threshold determined by the sum of the mean and standard deviation for the negative control (i.e., beads without antigen). **b**, **c** Volcano plots showing the results of multiple Wilcoxon matched pairs test that was FDR corrected for multiple comparisons (Benjamini) that compared each analyte **b** IgG vs IgM autoantibody levels or **c** IgG vs. IgA autoantibody levels. The dotted lines indicate FDR < 0.01. Analytes highlighted in red are significantly changed between antibody isotypes compared.

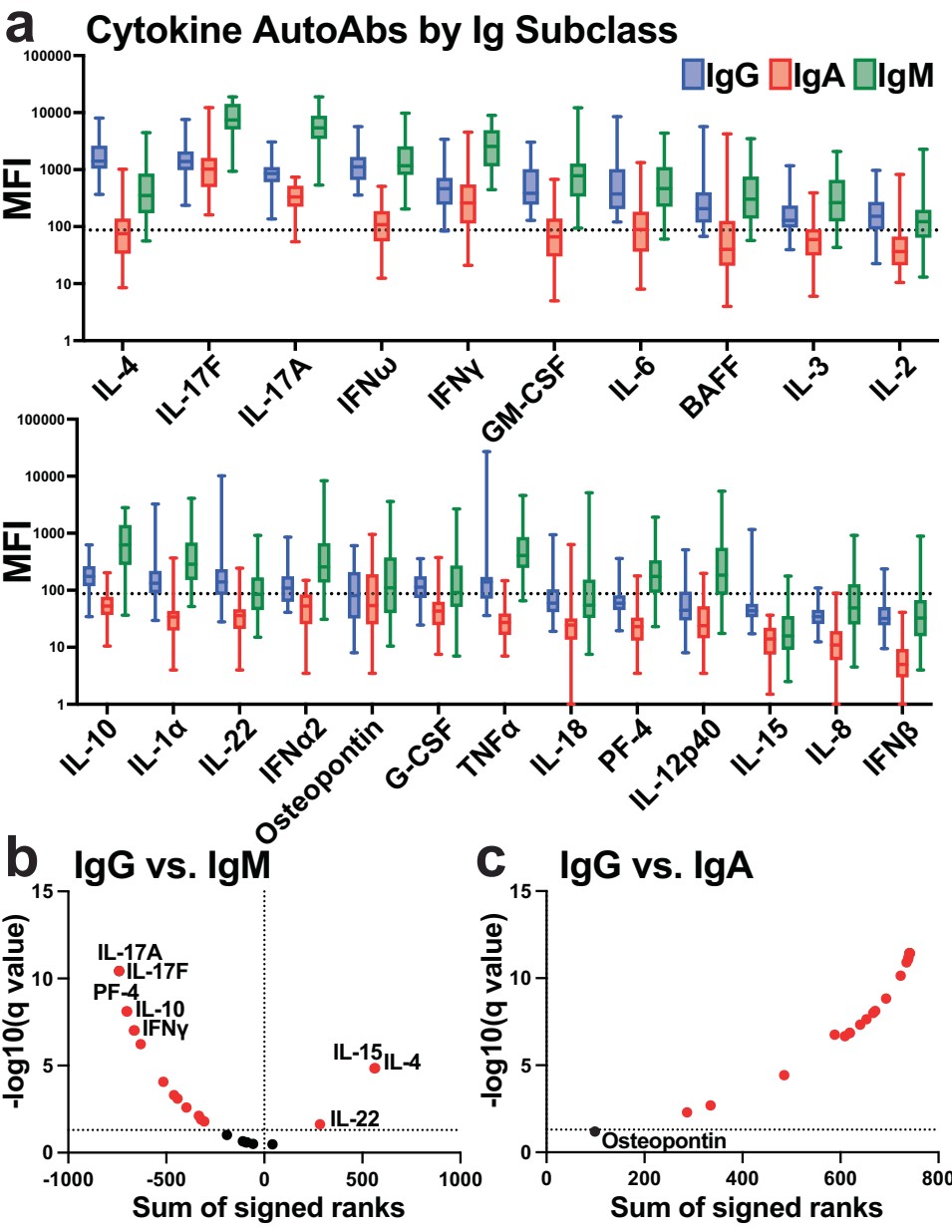

the first study demonstrating increased ACE2 antibody levels in IgG, IgA, and IgM based on disease severity[11,13,15,16]. Indeed, SARS-CoV-2 infection has been associated with increased ACE2 shedding from cell membranes, but there are conflicting studies on the impact of soluble ACE2 on enzymatic function[11,36–38]. Apart from its role as the SAR-CoV-2 receptor, membrane-bound ACE2 typically regulates blood pressure, wound healing, and inflammation via the renin–angiotensin system[6–8]. While we identified increased levels of ACE2 autoantibodies in individuals with severe COVID-19, future studies will be required to determine their impact on ACE2 function through isolation and characterization of ACE2-reactive B cells. Moreover, identification of other factors that could impact ACE2 activity during severe COVID-19 will also be required. By disrupting the balance of ACE2 through decreasing membrane-bound ACE2 and increasing autoantibodies binding to enzymatically active epitopes, enzymatic activity would decrease, and functions dependent on ACE2 regulation, such as inflammation, would be impacted and unregulated. Consistent with this observation, when ACE2 was knocked out in mice, proinflammatory cytokines were induced, and inflammation was increased in response to minor stress[9].

We used peptide microarray to map the high-resolution peptide binding of the ACE2 autoantibodies to the ACE2 protein. We found unique and common epitopes that were targeted by ACE2 autoantibodies in individuals with severe COVID-19. One limitation of this approach is that it does not identify structural non-linear conformational epitopes or epitopes that require post-translational protein modification and, therefore, may not provide all of the information relevant to ACE2 epitopes targeted by autoantibodies. None of the immunodominant epitopes were near the interaction site of ACE2 and SARS-CoV-2 RBD, and some targeted regions of the ACE2 protein have unknown roles in ACE2 function. Importantly, some of these binding sites were located within enzymatically active domains and could have an impact on ACE2 function. Future studies could determine if antibodies targeting these epitopes could block ACE2 activity and identify how B cells making these autoantibodies develop. Additionally, these antibody epitopes could be further explored as peptide biomarkers of COVID-19 disease severity.

Lastly, we found the presence of IgG, IgA, and IgM autoantibodies to a multitude of cytokines, chemokines, and other immune factors within healthy individuals and identified autoantibodies to immune factors that

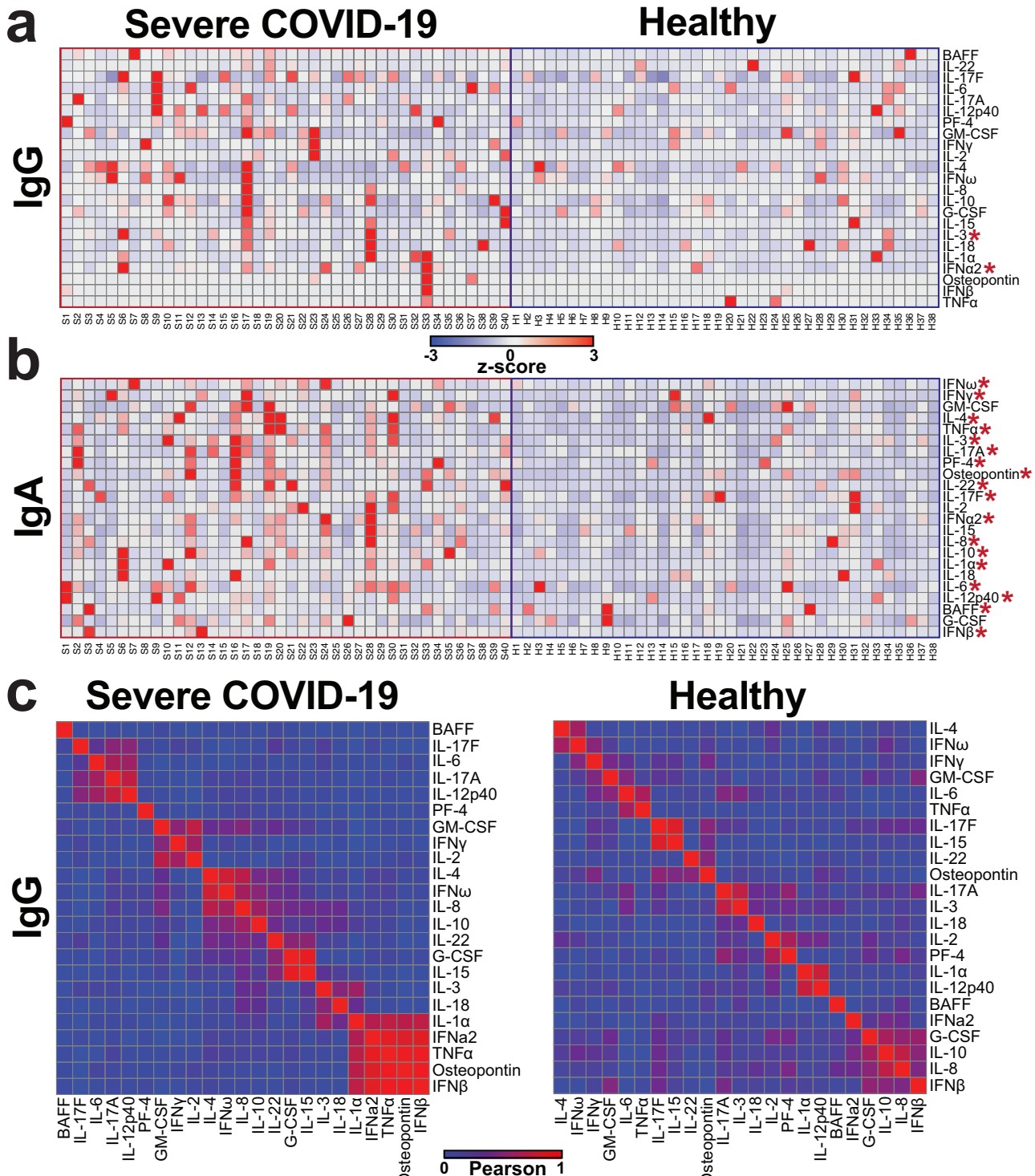

**Fig. 5 | Individuals with severe COVID-19 had significantly higher auto-antibodies to distinct analytes for IgG and IgA. a, b** Heatmaps of row z-scores of **a** IgG and **b** IgA autoantibody levels to the 23 cytokines, chemokines, and immune factors tested. Individuals with severe COVID-19 disease (left, *n* = 40) and healthy controls (right, *n* = 38). Analytes are significantly different in individuals with severe COVID-19 compared to controls are shown with a red asterisk. *, *p* ≤ 0.05; Wilcoxon–Mann–Whitney. **c** Pearson correlations of the IgG antibody levels between each analyte for severe COVID-19 individuals and healthy individuals.

were significantly higher in individuals with severe COVID-19. Even though autoantibodies exist at a basal level within healthy individuals, this study provides evidence that elevation of these autoantibodies, including ACE2 autoantibodies, is related to the severity of disease with exposure to SARS-CoV-2. This could indicate a natural immunoregulation system that involves antagonism of cytokines and other immune molecules through antibody-mediated mechanisms that have individual variation, even in healthy populations. This finding is consistent with previously published data, with reports of as much as 60% of hospitalized COVID-19 patients presenting with higher autoantibodies to certain antigens[12,14–16]. Patients with severe COVID-19 produced higher amounts of circulating neutrophils and increased neutrophil activation when compared to patients suffering

from milder forms of COVID-19[2,20]. Furthermore, patients infected with SARS-CoV-2 showed elevated levels of cytokines produced by neutrophils (IL-10, IL-8, and IL-6), which is consistent with our findings[2]. Wang et al. performed a large profiling for autoantibodies against 2770 proteins on a cohort of 194 individuals and discovered widespread autoantibody increases, with a striking enrichment in autoantibodies targeting interferons in patients with the most severe cases of COVID-19[15]. These researchers further demonstrated that pretreating transgenically expressed human ACE2 mice with neutralizing antibodies against interferon-alpha/beta receptor prior to SARS-CoV-2 infection led to increased disease severity and decreased survival[15]. Finally, over 10% of the patients with COVID-19 (from a large cohort of 987) had neutralizing IgG autoantibodies to type 1 interferons that could functionally neutralize type 1 interferons abilities to block SARS-CoV-2 infection in vitro[14]. Whether these autoantibodies target COVID-19-specific proteins or whether their upregulation is a more global response seen in multiple severe illnesses is still debatable[4,21,39,40]. Using a short longitudinal study, it was determined that a subset of autoantibodies were triggered specifically by SARS-COV-2 infection, but these patients were only followed for 7 days[12]. Interestingly, a recent study reported that 12 months after infection, patients with mild COVID-19 had higher autoantibody levels to cytokines such as CCL21, CXCL13, and CXCL16, compared to long COVID patients[39]. This finding is inconsistent with other reports, including a report examining immunological dysfunction 8 months after SARS-CoV-2 infection[4], but may suggest that certain autoantibodies could positively influence long term disease outcomes[39]. Whether the same autoantibodies are elevated during brief or prolonged time frames after infection is yet to be determined. Additionally, whether the same autoantibodies are triggered by other respiratory infections, for example, is still under investigation. A recent study examined IgG autoantibodies in 267 patients suffering from non-SARS-CoV-2 acute illnesses and discovered that autoantibodies are indeed upregulated in response to a wide variation of diseases and illnesses[21]. Specifically, anti-cytokine antibodies, such as TNFα, IL-2, IL-17A, and IFNα interferons, were found in all acute illnesses examined, with significantly higher levels in infected versus healthy individuals[21].

It is currently unclear whether all of the autoantibodies expressed at higher levels after COVID-19 disease are truly elevated in response to infection or whether some patients are predisposed to higher levels prior to infection, which in turn contribute to severe symptoms[41,42]. In the individuals with severe COVID-19, we found that the autoantibody levels of IL-1α, IFNα2, TNFα, osteopontin, and IFNβ showed a high correlation with each other. During COVID-19 disease, it has been previously documented that many cytokines, including IL-1α, IFNα, IFNβ, and TNFα, are involved in the transition from local to more systemic immune response to COVID-19 infection, increasing the likelihood of severe disease, sepsis, respiratory distress and possibility death[43–45]. Interestingly, recent studies have linked elevated osteopontin levels to COVID-19 disease severity, suggesting it could be used as a biomarker for COVID-19 severity[46,47]. Whether autoantibodies are the cause or the result of severe COVID-19, our study and others suggest that these autoantibodies may be used as predictive markers of disease severity in COVID-19 and other infections. Future work understanding how these autoantibodies develop and the mechanisms to block or enhance inflammation could uncover novel therapeutic approaches to controlling inflammation.

Our data demonstrated that SARS-CoV-2 infection can increase autoantibodies and show a correlation between elevated autoantibodies and COVID-19 disease severity. However, limitations were our study size, demographic makeup, and longitudinal follow-up. Moreover, determining if the autoantibodies truly contribute to immune regulation in vivo will be critical for determining the functional impact of this observation. Larger, longitudinal studies of diverse individuals will be required to fully characterize the fate and function of these autoantibodies to ACE2 and cytokines.

## Data availability
All data supporting the findings of this study are available with the paper and its supplementary information. The source data for the ACE2 peptide array is provided as Supplemental Data 1. Source data for other graphs can be found in Supplemental Data 2. Source data for the crystal structure of ACE2 in complex with the SARS-CoV-2 spike can be found in the protein data bank (PDB: 6MOJ)[23,25]. All other data are available from the corresponding author upon reasonable request.

## Code availability
This study did not generate any new software or custom code. GraphPad Prism version 9.1 (Boston, MA, USA) was used for graphing and statistical analysis. Heatmaps were generated using Morpheus (Broad Institute), and visualization of ACE2 and SARS-CoV-2 spike crystal structure and residue annotation was performed using EzMol version 2.1[24].

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

## Acknowledgements

We thank all the healthcare workers and donors who participated in this study. Special thanks to Occupational Health and Children's Mercy Research Institute for their support of this study. Daniel Louiselle, Nick Nolte, Rebecca Biswell, and Elizabeth Fraley assisted with peripheral blood sample processing and storage. Elin Grundberg, Tomi Pastinen, Bradley Beldon, Angela Myers, and Jennifer Schuster were involved in overseeing study design and clinical implementation with the mRNA vaccine. This research was supported by grants from the National Institutes of Health (NIH) (R01AI14778) and NIH funding from the Researching COVID-19 to Enhance Recovery (RECOVER) program of research.

## Author contributions

T.B. conceived and designed the study. E.S.G. and C.L. performed experiments. T.B. and E.S.G. performed data analysis. T.B. and E.S.G. interpreted the data. R.M., T.B. and E.S.G. wrote the paper, and all authors edited the paper.

## Competing interests

The authors declare no competing interests.
