## [Peer Review File · Communications Medicine]

Reviewers' comments:

Reviewer #1 (Remarks to the Author):

This is a well-written manuscript, whose results agree with the previously published results of Arthur et al (reference 11). I agree with the authors' results and conclusions. Line 106, "This" should be "These". I have no other recommendations.

Reviewer #2 (Remarks to the Author):

In this manuscript, autoantibodies to ACE2 and cytokines are examined in plasma from healthy donors and individuals 30-60 days after mild or severe COVID-19. The authors find significant increases in autoantibodies in the severe COVID-19 cohort and provide novel findings on the locations of epitopes in ACE2. While many aspects of this study have been previously investigated in other published works, considering that there is still much ambiguity and inconsistency in our understanding of molecular drivers of COVID-19 and PASC, I believe these kinds of studies are very important to add weight to specific hypotheses that may lead to successful medical interventions. Overall, the study is well described with high quality figures for presenting the data. The main limitations of the study are that absolute quantification of autoantibody levels is missing and it is not definitively shown that ACE2 autoantibodies inhibit ACE2 catalytic activity.

MAJOR COMMENT

While the authors show a correlation between severe COVID-19, ACE2 autoantibodies, and inhibition of ACE2 activity, it is not conclusively shown that the autoantibodies themselves are responsible for inhibiting ACE2. The kind of experiment that might convincingly show that autoantibodies are inhibitory would be to take some representative plasma samples with high inhibitory activity and deplete the plasma of antibodies (e.g. there are resins and magnetic beads commercially available that will bind and remove human IgM/IgA/IgG). If the authors' hypothesis is correct, immunoglobulin depletion would reduce the inhibitory activity of the plasma.

MINOR COMMENTS

1. In Figure 3B and in Supplementary Table S3, the authors use peptide # (from 1 to 199) instead of amino acid residue number. Peptide # is meaningless to anybody but the authors, whereas ACE2 residue number is universally understood. Please convert the x-axis of Figure 3B and the relevant column of Table S3 to actual amino acid numbers of ACE2.
2. Please clarify in the legend of Figure 3 that epitope mapping is specifically for IgG isotype.
3. When the peptides listed in Table S3 are mapped to the structure of ACE2 (PDB 6M17), it is found that the epitopes are primarily localized to the *backside* of the ACE2 protease domain where they are exposed to solvent (i.e. not in the enzyme active site). The authors should add a supplementary figure showing where the epitopes map on the ACE2 structure and tone down their language that epitope mapping supports inhibition of the enzyme active site.
4. The epitope mapping strategy used by the authors only identifies linear epitopes in the ACE2

polypeptide. Structural epitopes are not identified. The technical approach the authors have taken is still very informative but I recommend they add a cautionary statement that their method is blind to structural epitopes and therefore only provides a partial picture of ACE2 epitopes.

5. The ELISA results are presented as OD450 rather than absolute quantification (i.e. ng/ml or nM) and OD450 values may not reflect linear differences in antigen concentration. While disappointing, from a practical perspective I sympathize with why the authors chose to only present OD450. Absolute quantification is much harder and much more labor intensive, as it requires having excellent standard curves and evaluating samples at multiple dilutions to obtain OD values that fit in the linear region of the standard curves. If the authors are unable to get absolute quantities for autoantibody levels, then they should at least make it clear to readers that their data shows trends but not absolute differences or even absolute relative differences.

6. Is there a correlation in post-severe COVID-19 individuals between IgM, IgA, and IgG autoantibodies? Or can individuals have high autoantibodies of one isotype and low for a different isotype?

7. The authors write, "The highest overall levels of ACE2 autoantibodies were observed for IgA isotype, followed by IgM and IgG." But without standards and absolute quantitation, do the authors really know that a higher OD450 in an IgA ELISA means more antigen than a lower OD450 in an IgM or IgG ELISA?

8. The authors write, "This could indicate a natural immunoregulation system that involves antagonism of cytokines and other immune molecules through antibody-mediated mechanisms, that has individual variation, even in healthy populations." I am skeptical and it may be just a reflection of partial tolerance mechanisms. I would suggest moving this sentence to later paragraphs in the paper so that it is clear this is a speculative discussion point.

9. The following sentence is confusing and should be edited to make it clear they are referring to levels of autoantibodies versus expression levels of cytokines:
In the severe COVID-19 group, IL-1a, IFN α 2, TNF α , osteopontin and IFN β were one of the largest groups of analytes that had correlated expression levels.

10. Likewise, the following sentence is also confusing and makes it seem like the autoantibodies are towards immunoglobulin isotypes:
Moreover, individuals with severe COVID-19 had significantly higher levels of autoantibodies to these isotypes for both IgG and IgA, but not IgM.

11. In the legend to Figure 4, the use of the terms "IgG vs IgM expression levels" and "IgG vs IgA expression levels" is a bit confusing. Why not use instead "autoantibody levels" instead of "expression levels"?

12. The authors write, "Indeed, SARS-CoV-2 infection has been associated with increased ACE2 shedding from cell membranes and increased plasma ACE2 catalytic activity." While there is definitely increased soluble ACE2 fragments in serum, studies disagree on whether there is increased catalytic activity. For example, see the following reports:
[https://www.cell.com/molecular-therapy-family/methods/fulltext/S2329-0501\(22\)00095-X](https://www.cell.com/molecular-therapy-family/methods/fulltext/S2329-0501(22)00095-X)
<https://www.ncbi.nlm.nih.gov/pmc/articles/PMC9429950/>

13. In Figure 5C, does the group of IL-1a, IFN α 2, TNF α , osteopontin and IFN β have common physiology compared to the other cytokines? Which cytokines are published as having elevated expression in COVID and thus may induce autoantibodies? Anything interesting worth adding to their manuscript?

Erik Procko

Reviewer #3 (Remarks to the Author):

This is a well-written manuscript detailing the presence of auto-antibodies to ACE2 receptor and important inflammatory cytokines and chemokines. The authors go one step further to evaluate the function of the autoantibodies against ACE2R in the serum of patients with severe COVID-19. These antibodies alter function dramatically. They also map the epitopes to show where each antibody binds. The authors show that severe COVID-19 causes increased levels of autoantibodies to ACE2R, cytokines, and chemokines. The authors mention that these antibodies may effect long-term outcomes, but this language should be tempered as this finding was not the outcome for which this study was powered. Otherwise, this reviewer would accept this manuscript.

Minor:

Temper language around PASC and the autoantibodies as this is not what this study was powered to evaluate.

We thank the reviewers for their time and consideration in reviewing our manuscript. With this revision, we have addressed each of their comments and suggestions. We have also revised the formatting of the manuscript to meet the publication guidelines of *Communications Medicine*. Below is a point-by-point response to each reviewer comment. We believe this has greatly improved our study resulting in a stronger manuscript with increased clarity and scientific precision.

Reviewer comments

Reviewer #1 (Remarks to the Author):

This is a well-written manuscript, whose results agree with the previously published results of Arthur et al (reference 11). I agree with the authors' results and conclusions. Line 106, "This" should be "These". I have no other recommendations.

RESPONSE: We thank the reviewer for their positive feedback and time for reviewing the manuscript. "This" has been changed to "These" on line 106.

Reviewer #2 (Remarks to the Author):

In this manuscript, autoantibodies to ACE2 and cytokines are examined in plasma from healthy donors and individuals 30-60 days after mild or severe COVID-19. The authors find significant increases in autoantibodies in the severe COVID-19 cohort and provide novel findings on the locations of epitopes in ACE2. While many aspects of this study have been previously investigated in other published works, considering that there is still much ambiguity and inconsistency in our understanding of molecular drivers of COVID-19 and PASC, I believe these kinds of studies are very important to add weight to specific hypotheses that may lead to successful medical interventions. Overall, the study is well described with high quality figures for presenting the data. The main limitations of the study are that absolute quantification of autoantibody levels is missing and it is not definitively shown that ACE2 autoantibodies inhibit ACE2 catalytic activity.

RESPONSE: We thank the reviewer for their time and critiques. We agree with the positive comments and the importance of the study. We have addressed the limitations raised in the specific points below.

MAJOR COMMENT

While the authors show a correlation between severe COVID-19, ACE2 autoantibodies, and inhibition of ACE2 activity, it is not conclusively shown that the autoantibodies themselves are responsible for inhibiting ACE2. The kind of experiment that might convincingly show that autoantibodies are inhibitory would be to take some representative plasma samples with high inhibitory activity and deplete the plasma of antibodies (e.g. there are resins and magnetic beads commercially available that will bind and remove human IgM/IgA/IgG). If the authors' hypothesis is correct, immunoglobulin depletion would reduce the inhibitory activity of the plasma.

RESPONSE: We agree with the reviewer that there could be other factors in the plasma of individuals with severe COVID-19 that could reduce ACE2 function. Unfortunately, we do not have ample plasma samples to comprehensively deplete all isotypes of Ig and retest. Even then, we believe that B cells that are reactive for ACE2 would need to be isolated and expressed to study function in future experiments to define how ACE2 autoantibodies may regulate ACE2 function. Thus, to avoid any speculation, we have removed the figure that was describing the inhibition assay. We have also removed the associated text describing impact on ACE2 function.

We have added text to the discussion section that highlights this point "While we identified increased levels of ACE2 autoantibodies in individuals with severe COVID-19, future studies will be required to determine their

impact on ACE2 function through isolation and characterization of ACE2-reactive B cells. Moreover, identification of other factors that could impact ACE2 activity during severe COVID-19 will also be required.”

MINOR COMMENTS

1. In Figure 3B and in Supplementary Table S3, the authors use peptide # (from 1 to 199) instead of amino acid residue number. Peptide # is meaningless to anybody but the authors, whereas ACE2 residue number is universally understood. Please convert the x-axis of Figure 3B and the relevant column of Table S3 to actual amino acid numbers of ACE2.

RESPONSE: We thank the reviewer for pointing this out. The peptides are 15 amino acid long, but overlap with each other by 11 amino acids, this overlap makes labeling by ACE2 amino acid position alone challenging over the span of ACE2. We have now added two additional columns to Table S2 that has the ACE2 amino acid start and end position for each peptide. Additionally, the entire peptide amino acid sequence is provided in the table for reference. We hope the inclusion of the ACE2 peptide sequence and amino acid position will clarify the position on the ACE2 protein.

2. Please clarify in the legend of Figure 3 that epitope mapping is specifically for IgG isotype.

RESPONSE: We have clarified this in the figure legend.

3. When the peptides listed in Table S3 are mapped to the structure of ACE2 (PDB 6M17), it is found that the epitopes are primarily localized to the *backside* of the ACE2 protease domain where they are exposed to solvent (i.e. not in the enzyme active site). The authors should add a supplementary figure showing where the epitopes map on the ACE2 structure and tone down their language that epitope mapping supports inhibition of the enzyme active site.

RESPONSE: We thank the reviewer for this comment and suggestion for further structural analysis of the linear epitopes within ACE2. We have mapped the linear epitopes using a crystal structure of ACE2 in complex with the SARS-CoV-2 receptor-binding domain (PDB 6M0J). As the reviewer pointed out, none of the linear antibody epitopes were near the SARS-CoV-2 RBD interaction with ACE2. Four other epitopes did map to sites not within the enzymatic site with unknown function. However, using structural analysis of an inhibitor of ACE2 substrate binding and function (MLN-4760; Towler et al. J. Biol. Chem. 2004), we also mapped residues important for interaction with the ACE2 inhibitor. These 18 residues are within the cleft of ACE2 that is important for catalytic activity. We found that two of the linear epitopes were overlapping with the same region that the ACE2 inhibitor interacts with ACE2. Thus, these two epitopes should be further studied in their role of ACE2 function. We have included these results as a new Figure 3 (which includes moving the supplementary table to the new figure) that describes these findings. We have also toned-down language that the epitope mapping entirely supports inhibition of the enzyme active site.

We have added the following the results section:

“We further analyzed the location of the immunodominant linear antibody epitopes on the crystal structure of ACE2 in complex with the SAR-CoV-2 receptor-binding domain (RBD; PDB:6M0J). The SARS-CoV-2 spike RBD binding to the ACE2 protein is independent of ACE2 catalytic activity and occurs on the outer surface of the molecule (Figure 3B). One of the peptide epitopes (peptide ID: g) was in the C-terminal collectrin-like domain and not resolved in this crystal structure. Four other peptide regions (peptide ID: b, d, e, f) bind to the outside of the ACE2 protein, but not near the SARS-CoV-2 RBD interaction site or the cleft important for ACE2 catalytic activity (Figure 3B). Two of the immunodominant peptide epitopes are located in the ACE2 cleft important for substrate binding and enzymatic activity (peptide ID: a, c; Figure 3B). Prior structural analysis of an ACE2 catalytic inhibitor (MLN-4760) identified key residues important for ACE2 substrate binding and carboxypeptidase activity. The two peptide regions targeted by antibodies in multiple individuals

(peptide id: a, c) overlap with these regions previously shown to be important for ACE2 substrate binding and subsequent enzyme activity (Figure 3C). These data showed that ACE2 autoantibodies had common linear epitopes across many individuals and some antibody epitopes targeted regions that may be important for ACE2 enzymatic activity.”

We have modified the discussion section:

“We used peptide microarray to map the high-resolution peptide binding of the ACE2 autoantibodies to the ACE2 protein. We found unique and common epitopes that were targeted by ACE2 autoantibodies in individuals with severe COVID-19. One limitation of this approach is that it does not identify structural non-linear conformational epitopes, or epitopes that require post-translational protein modification, and therefore may not provide all of the information relevant to ACE2 epitopes targeted by autoantibodies. None of the immunodominant epitopes were near the interaction site of ACE2 and SARS-CoV-2 RBD, and some targeted regions of the ACE2 protein that have unknown roles in ACE2 function. Importantly, some of these binding sites were located within enzymatically active domains and could have an impact ACE2 function. Future studies could determine if antibodies targeting these epitopes could block ACE2 activity and identify how B cells making these autoantibodies develop. Additionally, these antibody epitopes could be further explored as peptide biomarkers of COVID-19 disease severity.”

4. The epitope mapping strategy used by the authors only identifies linear epitopes in the ACE2 polypeptide. Structural epitopes are not identified. The technical approach the authors have taken is still very informative but I recommend they add a cautionary statement that their method is blind to structural epitopes and therefore only provides a partial picture of ACE2 epitopes.

RESPONSE: We have added a sentence into the discussion paragraphs addressing this caveat.

“One limitation of this approach is that it does not identify structural non-linear conformational epitopes, or epitopes that require post-translational protein modification, and therefore may not provide all of the information relevant to ACE2 epitopes targeted by autoantibodies.”

5. The ELISA results are presented as OD450 rather than absolute quantification (i.e. ng/ml or nM) and OD450 values may not reflect linear differences in antigen concentration. While disappointing, from a practical perspective I sympathize with why the authors chose to only present OD450. Absolute quantification is much harder and much more labor intensive, as it requires having excellent standard curves and evaluating samples at multiple dilutions to obtain OD values that fit in the linear region of the standard curves. If the authors are unable to get absolute quantities for autoantibody levels, then they should at least make it clear to readers that their data shows trends but not absolute differences or even absolute relative differences.

RESPONSE: Yes, we agree with the reviewer. We did not have access to an ACE2 antibody in all three isotypes that could be utilized as a standard for absolute quantification. Hopefully in future studies this could be incorporated. Even if we had these standards the primary antibodies used for the different Ig subclasses would be different and thus, could have differences in efficiency of determining comparable absolute quantification. Although samples were run in parallel between immunoglobulin types for ELISA, we agree quantification of antibodies does present superior confidence in measured antibody levels and for comparisons. However, the results of other publications that report ACE2 antibodies or other autoantibodies in OD450 formatting for ELISA (Casciola-Rosen et al 2022 JCI Insight) or fluorescence intensity for bead based assays (Bastard, P. et al 2020 Science, Chang, S. E. et al. 2021 Nat Comm) instead of absolute concentration, and have similar challenges.

We have modified the text of the results section to make this clear to the readers:

“First, we used plasma samples from healthy individuals and determined individuals that were considered positive for ACE2 autoantibodies using a positivity cut-off of twice the background reading for each Ig isotype (Supplementary Figure 1). We did not have access to isotype-specific ACE2 autoantibody standards so the levels are reported are comparing intensity of the colorimetric substrate measured as optical density. The highest overall OD450 levels of ACE2 autoantibodies were observed for IgA isotype, followed by IgM and IgG.

6. Is there a correlation in post-severe COVID-19 individuals between IgM, IgA, and IgG autoantibodies? Or can individuals have high autoantibodies of one isotype and low for a different isotype?

RESPONSE: Based on Figure 1 where the dots present individuals, almost every severe COVID-19 individual had IgG, IgA and IgM autoantibodies that were higher than the means for mild COVID-19 and healthy. We generated a heatmap for each individuals and each Ig subclass of ACE2 autoantibodies. Using Pearson correlation of z-scores of immunoglobulin ELISA OD450 values, there was a significant correlation between samples for IgA and IgM ($p= 0.0391$), but no correlation between IgG vs IgA, or IgG vs IgM. Thus, some individuals had high levels of all or multiple Ig subclasses of ACE2 autoantibodies but not all. We have added the heatmap and correlation analysis to supplemental figure 2. We have added the following statement to the results describing this figure.

“We also found a significant correlation between IgA and IgM ACE2 autoantibodies in the individuals with severe COVID-19, but not with IgG and IgA or IgM subclasses (Supplemental Figure 2).”

7. The authors write, "The highest overall levels of ACE2 autoantibodies were observed for IgA isotype, followed by IgM and IgG." But without standards and absolute quantitation, do the authors really know that a higher OD450 in an IgA ELISA means more antigen than a lower OD450 in an IgM or IgG ELISA?

RESPONSE: We have modified this section in response to comment #5.

8. The authors write, "This could indicate a natural immunoregulation system that involves antagonism of cytokines and other immune molecules through antibody-mediated mechanisms, that has individual variation, even in healthy populations." I am skeptical and it may be just a reflection of partial tolerance mechanisms. I would suggest moving this sentence to later paragraphs in the paper so that it is clear this is a speculative discussion point.

RESPONSE: We have moved this sentence into the discussion paragraphs.

9. The following sentence is confusing and should be edited to make it clear they are referring to levels of autoantibodies versus expression levels of cytokines:

In the severe COVID-19 group, IL-1 α , IFN α 2, TNF α , osteopontin and IFN β were one of the largest groups of analytes that had correlated expression levels.

RESPONSE: We have reworded this sentence to read “In the severe COVID-19 group, levels of autoantibodies against IL-1 α , IFN α 2, TNF α , osteopontin and IFN β all showed high correlation”

10. Likewise, the following sentence is also confusing and makes it seem like the autoantibodies are towards immunoglobulin isotypes:

Moreover, individuals with severe COVID-19 had significantly higher levels of autoantibodies to these isotypes for both IgG and IgA, but not IgM.

RESPONSE: We have reworded this sentence to read “Moreover, individuals with severe COVID-19 had significantly higher levels of autoantibodies targeting cytokines and immune factors for both IgG and IgA, but not IgM, immunoglobulin subclasses.”

11. In the legend to Figure 4, the use of the terms "IgG vs IgM expression levels" and "IgG vs IgA expression levels" is a bit confusing. Why not use instead "autoantibody levels" instead of "expression levels"?

RESPONSE: Great suggestion and we have changed the figure legend accordingly.

12. The authors write, "Indeed, SARS-CoV-2 infection has been associated with increased ACE2 shedding from cell membranes and increased plasma ACE2 catalytic activity." While there is definitely increased soluble ACE2 fragments in serum, studies disagree on whether there is increased catalytic activity. For example, see the following reports:

[https://www.cell.com/molecular-therapy-family/methods/fulltext/S2329-0501\(22\)00095-X](https://www.cell.com/molecular-therapy-family/methods/fulltext/S2329-0501(22)00095-X)

<https://www.ncbi.nlm.nih.gov/pmc/articles/PMC9429950/>

RESPONSE: We have added the suggested references and modified the text of this statement to reflect the conflicting studies.

“Indeed, SARS-CoV-2 infection has been associated with increased ACE2 shedding from cell membranes, but there are conflicting studies on the impact of soluble ACE2 on enzymatic function.”

13. In Figure 5C, does the group of IL-1a, IFN α 2, TNF α , osteopontin and IFN β have common physiology compared to the other cytokines? Which cytokines are published as having elevated expression in COVID and thus may induce autoantibodies? Anything interesting worth adding to their manuscript?

RESPONSE: We thank the reviewer for highlighting this important observation and have expanded on that is currently know about these cytokines with respect to COVID-19 severity. Interestingly, osteopontin has already been suggested as a biomarker for COVID-19. We have added the following information, with references, to the discussion section of the manuscript.

“In the individuals with severe COVID-19, we found that the autoantibody levels to IL-1 α , IFN α 2, TNF α , osteopontin and IFN β showed high correlation with each other. During COVID-19 disease, it has been previously documented that many cytokines, including IL-1 α , IFN α , IFN β and TNF α , are involved in the transition from local to more systemic immune response to COVID-19 infection, increasing the likelihood of severe disease, sepsis, respiratory distress and possibility death. Interestingly, recent studies have linked elevated osteopontin levels to COVID-19 disease severity suggesting it could be used as a biomarker for COVID-19 severity.”

Reviewer #3 (Remarks to the Author):

This is a well-written manuscript detailing the presence of auto-antibodies to ACE2 receptor and important inflammatory cytokines and chemokines. The authors go one step further to evaluate the function of the autoantibodies against ACE2R in the serum of patients with severe COVID-19. These antibodies alter function dramatically. They also map the epitopes to show where each antibody binds. The authors show that severe COVID-19 causes increased levels of autoantibodies to ACE2R, cytokines, and chemokines. The authors mention that these antibodies may effect long-term outcomes, but this language should be tempered as this finding was not the outcome for which this study was powered. Otherwise, this reviewer would accept this manuscript.

Minor:

Temper language around PASC and the autoantibodies as this is not what this study was powered to evaluate.

RESPONSE: We thank the reviewer for this comment and agree that our study was not powered or have the correct human subject cohort to answer this question. We have removed any reference to PASC.

REVIEWERS' COMMENTS:

Reviewer #1 (Remarks to the Author):

The authors has satisfactorily revised the manuscript.

Reviewer #2 (Remarks to the Author):

Overall, the revisions improve the manuscript, but I still don't understand why the authors want to keep using peptide # in their revised figures, especially in Figure 3A. The heading for column 3 in Figure 3A is "ACE2 amino acid sequence number" but what they provide is a peptide # ID. Why not actually give the amino acid sequence number as universally understood by the scientific community? This is a very easy fix:

Peptide ID	Peptide Sequence	ACE2 a.a. number
a (48, 49)	EMARANHYEDYGDYWRGDY	189-207
b (58)	TFEEIKPLYEHLHAY	229-243
c (64-67)	PSYISPIGCLPAHLLGDMWGRFWTNLY	253-279

etc.

Finally, the authors may find the following publication interesting for their future research on this topic, since it investigates whether ACE2 autoantibodies are inhibitory:

<https://pubmed.ncbi.nlm.nih.gov/36856018/>

DOI: 10.1002/eji.202250210

Reviewer #3 (Remarks to the Author):

The authors have been very responsive to the reviewers' comments. The overall manuscript has been improved dramatically. At this time, this reviewer does not find any other issues that should prevent the publication of this manuscript.

Author response to reviewers.

We thank all the reviewers for their time and attention to detail.

Reviewer #1 (Remarks to the Author):

The authors has satisfactorily revised the manuscript.

Reviewer #2 (Remarks to the Author):

Overall, the revisions improve the manuscript, but I still don't understand why the authors want to keep using peptide # in their revised figures, especially in Figure 3A.

Comment 2-1 The heading for column 3 in Figure 3A is "ACE2 amino acid sequence number" but what they provide is a peptide # ID. Why not actually give the amino acid sequence number as universally understood by the scientific community? This is a very easy fix:

Peptide ID Peptide Sequence ACE2 a.a. number
a (48, 49) EMARANHYEDYGDYWRGDY 189-207
b (58) TFEEIKPLYEHLHAY 229-243
c (64-67) PSYISPIGCLPAHLLGDMWGRFWTNLY 253-279
etc.

Response: We have updated Table 1 (formerly Figure 3A) with this information.

Comment 2-2 Finally, the authors may find the following publication interesting for their future research on this topic, since it investigates whether ACE2 autoantibodies are inhibitory:

<https://pubmed.ncbi.nlm.nih.gov/36856018/>

DOI: 10.1002/eji.202250210

Response: We thank the reviewer for finding this publication and have added this reference into the discussion.

Reviewer #3 (Remarks to the Author):

The authors have been very responsive to the reviewers' comments. The overall manuscript has been improved dramatically. At this time, this reviewer does not find any other issues that should prevent the publication of this manuscript.